# Using both qualitative and quantitative data in parameter identification for systems biology models

Eshan D. Mitra [1], Raquel Dias [2,3], Richard G. Posner [2] & William S. Hlavacek [1]

In systems biology, qualitative data are often generated, but rarely used to parameterize models. We demonstrate an approach in which qualitative and quantitative data can be combined for parameter identification. In this approach, qualitative data are converted into inequality constraints imposed on the outputs of the model. These inequalities are used along with quantitative data points to construct a single scalar objective function that accounts for both datasets. To illustrate the approach, we estimate parameters for a simple model describing Raf activation. We then apply the technique to a more elaborate model characterizing cell cycle regulation in yeast. We incorporate both quantitative time courses (561 data points) and qualitative phenotypes of 119 mutant yeast strains (1647 inequalities) to perform automated identification of 153 model parameters. We quantify parameter uncertainty using a profile likelihood approach. Our results indicate the value of combining qualitative and quantitative data to parameterize systems biology models.

[1] Theoretical Biology and Biophysics Group, Theoretical Division, Los Alamos National Laboratory, Los Alamos, NM 87545, USA. [2] Department of Biological Sciences, Northern Arizona University, Flagstaff, AZ 86011, USA. [3] Present address: The Scripps Research Institute, La Jolla, CA 92037, USA. Correspondence and requests for materials should be addressed to W.S.H. (email: wish@lanl.gov)

Systems biology models, such as those found in BioModels Database[1], typically have outputs in the form of time courses. It follows that if one wants to parameterize such a model, the most useful dataset would be the corresponding experimental time courses. However, time-course data may be unavailable, limited, or corrupted by noise.

Much of the experimental data collected in biology are qualitative, categorical characterizations, such as activating or repressing, oscillatory or non-oscillatory, or lower or higher relative to a control. In contrast, quantitative data are numerical and may take the form of a time course, a steady-state dose-response curve, a distribution, or a ratio.

Although qualitative observations are not numerical, they still contain information that could potentially aid in model development. However, qualitative observations are largely ignored by the modeling community.

A notable exception is the modeling work of Tyson and co-workers on the cell cycle in budding yeast[2–6]. In ref. [3], parameters were tuned using qualitative data: the viability or inviability of 131 yeast mutant strains. Parameters in yeast cell cycle models have been estimated by hand-tuning[3] and later refined by automated tuning[7] to maximize the number of mutant strains that the models describe correctly.

Here, we extend the approach of Oguz et al.[7], and demonstrate how qualitative biological observations can be formalized as inequality constraints on the outputs of a model. Such a formulation has the advantages that (1) it is generalizable to a range of biological problems, whenever qualitative data are available; and (2) it lends itself to the use of quantitative data in addition to the qualitative data in parameter identification. Constrained optimization—the task of minimizing an objective function subject to inequality constraints—is well-studied in the field of optimization[8]. In the context of parameter identification for a biological model, we minimize the sum-of-squares distance from the quantitative data, and each qualitative data point leads to one inequality constraint.

Constrained optimization can also be viewed as an extension of model checking[9], a technique with applications in systems biology[10,11]. Model checking seeks to verify that a model meets a set of desired specifications. Here, we consider specifications that are straightforward to verify for a single model (inequality constraints on outputs of deterministic models), but we seek to tune model parameters to achieve optimal agreement with the specifications.

Numerous algorithms for constrained optimization are known[8]. Because the constraints in biological modeling are derived from experimental data, they have some level of uncertainty, and it may be reasonable to tolerate parameterizations for which some constraints are not satisfied. Static penalty functions appropriately handle these soft constraints by adding to the objective function a cost proportional to the extent of each constraint violation[12]. This converts a constrained optimization problem to minimization of a scalar function, which can be approached using, for example, a metaheuristic optimization method[13].

To illustrate the strengths of constrained optimization for biological modeling, we consider the budding yeast model of ref. [7]. We formulate the associated yeast phenotypic data in terms of inequality constraints, and perform automated parameter identification. We incorporate quantitative data that has not been used previously to parameterize this model. We also account for data about the phase of cell cycle arrest in inviable yeast mutants —additional qualitative data that previously required hand-tuning of parameters to incorporate[6]. Uncertainty quantification shows that the combination of quantitative and qualitative data leads to a higher level of confidence in parameter estimates than either dataset individually.

## Results

**An illustration of the potential value of qualitative data.** To demonstrate the potential value of qualitative data, we consider a simple case of solving for the coefficients of polynomial functions.

We consider two polynomial functions: $y_1 = ax^2 - bx + c$ and $y_2 = dx + e$. Suppose we want to solve for the coefficients $a$, $b$, $c$, $d$, and $e$, which we will take to be positive. As the ground truth coefficients to be determined, we choose $(a, b, c, d, e) = (0.5, 3, 5, 1, 1.5)$.

Suppose that a limited amount of quantitative information is available. Namely, it is known that the parabola $y_1$ contains the points (2, 1) and (8, 13), and the line $y_2$ contains the point (3.5, 5). This is not enough information to solve for any of the coefficients because three points are required to specify a parabola, and two points are required to specify a line (Fig. 1a).

Suppose that in addition, there is qualitative information: at some number of $x$ values, we know whether $y_2 > y_1$ (+) or $y_2 < y_1$ (−) (Fig. 1a). This information constrains the $x$ values at which $y_1$ and $y_2$ intersect. For example, with qualitative information at five points as shown in Fig. 1a, we know that the two functions intersect at a value $0 < x_1 < 2.5$ and at a value $5 < x_2 < 7.5$. Equivalently, these are bounds on the roots of the parabola $y_1 - y_2 = ax^2 - (b + d)x + (c - e)$.

Note that if the roots $x_1$ and $x_2$ of $y_1 - y_2$ are known exactly, then we can obtain exact solutions for all five coefficients. By Vieta's formulas for the sum and product of roots, we have $x_1 + x_2 = (b + d)/a$ and $x_1 x_2 = (c - e)/a$. Combined with the three known quantitative points, we have a solvable system of five linear equations.

If we have bounds such as $0 < x_1 < 2.5$ and $5 < x_2 < 7.5$, how well can we estimate the coefficients? To address this question, we sampled possible values of $x_1$ and $x_2$ within the bounds, solved the resulting linear equations, and, among solutions with all parameters positive, determined the range of values that each coefficient could take.

As the number of known qualitative points increases (assumed to be uniformly distributed over the range $0 \leq x \leq 10$), the bounds on $x_1$ and $x_2$ become tighter, leading to tighter bounds on the values of the coefficients (Fig. 1b). With a sufficient amount of

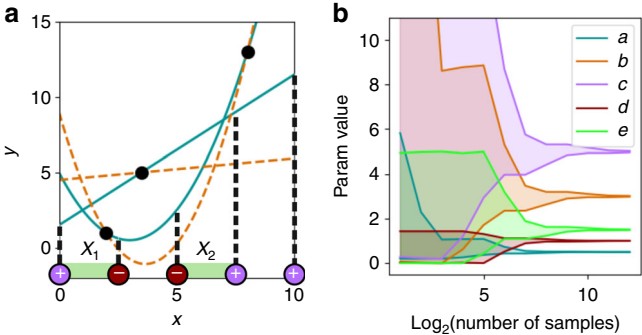

**Fig. 1** A simple illustration using polynomial functions. We use qualitative and quantitative information to determine the unknown coefficients. **a** Visualization of the problem. We seek to find the coefficients of equations for a parabola and a line, with the ground truth shown (blue solid curves). Two points on the parabola and one point on the line are known (black dots). These three points are consistent with infinitely many possible solutions (e.g., orange dashed curves). Qualitative information (colored circles, x-axis) specifies whether the parabola is above (+) or below (−) the line. This information limits the possible values of intersection points $x_1$ and $x_2$ to the green shaded segments of the x-axis. **b** Bounds on coefficient values as a function of the number of qualitative points known. Shaded areas indicate the range of possible values of each coefficient

qualitative information, the coefficient estimates converge to the ground truth values.

This example illustrates the potential utility of qualitative data in identifying model parameter values.

**Fitting with qualitative and quantitative data.** In the following sections, we will demonstrate an approach by which we incorporate both qualitative and quantitative data into a parameter identification procedure. We minimize an objective function with contributions from both types of data:

$$f_{tot}(\mathbf{x}) = f_{quant}(\mathbf{x}) + f_{qual}(\mathbf{x}) \tag{1}$$

where $\mathbf{x}$ is the vector of the unknown model parameters. The term $f_{quant}(\mathbf{x})$ is a standard sum of squares over all quantitative data points $j$:

$$f_{quant}(\mathbf{x}) = \sum_j \left( y_{j,model}(\mathbf{x}) - y_{j,data} \right)^2 \tag{2}$$

For the term $f_{qual}(\mathbf{x})$, we construct a function that likewise takes a lower value as the qualitative data are better matched by model outputs. We express each qualitative data point as an inequality of the form $g_i(\mathbf{x}) < 0$. Given this constraint, we seek to minimize the value of $C_i \cdot \max(0, g_i(\mathbf{x}))$, where $C_i$ is a problem-specific constant. In other words, if the constraint $g_i(\mathbf{x}) < 0$ is violated, we apply a penalty proportional to the magnitude of constraint violation. The final objective function to be minimized consists of the sum of the penalties arising from all of the individual constraints:

$$f_{qual}(\mathbf{x}) = \sum_i C_i \cdot \max(0, g_i(\mathbf{x})) \tag{3}$$

In the constrained optimization literature, Eq. (3) is called a static penalty function[12]. The squared difference, $\max(0, g_i(\mathbf{x}))^2$, is also sometimes used, but we chose Eq. (3) to avoid overpenalizing single constraints with a large degree of violation.

$f_{tot}(\mathbf{x})$ can be minimized using a standard optimization algorithm such as differential evolution[14] or scatter search[15].

**Fitting a model of Raf inhibition.** Next, we demonstrate usage of a combination of qualitative and quantitative data for parameter identification (which we will also refer to as finding a fit to the data) for a simple biological model. We use synthetic data to parameterize a model, originally described in Kholodenko et al.[16], for the dimerization of the protein kinase Raf and the inhibition of Raf by a kinase inhibitor. This model has relevance for cancer treatment[17,18]. We show that qualitative data can be used to improve confidence limits on the parameter values.

We consider the model shown in Fig. 2a. Raf (R) is able to dimerize, and each Raf monomer is able to bind an inhibitor (I). The model parameters consist of six equilibrium constants, denoted $K_1$ through $K_6$. Note that these six parameters are not independent. By detailed balance, $K_4 = K_1 K_3 / K_2$, and $K_6 = K_4 K_5 / K_2 = K_1 K_3 K_5 / K_2^2$, leaving four independent model parameters.

For the purposes of this example, we assume that $K_1$ and $K_2$ are known. $K_1$ could be obtained by studying the dimerization of Raf in the absence of inhibitor, and $K_2$ could be obtained by studying inhibitor binding to a dimerization-incompetent Raf mutant. Our problem of interest, therefore, is solving for the parameters $K_3$ and $K_5$.

The model has six population variables, which we denote as $R$, $I$, RR, RI, RIR, and RIRI. It is also useful to consider the total abundance of Raf, $R_{tot} = R + RI + 2(RR + RIR + RIRI)$. Two population variables, e.g. $R_{tot}$ and $I$, must be specified to define the entire system state given the equilibrium constants.

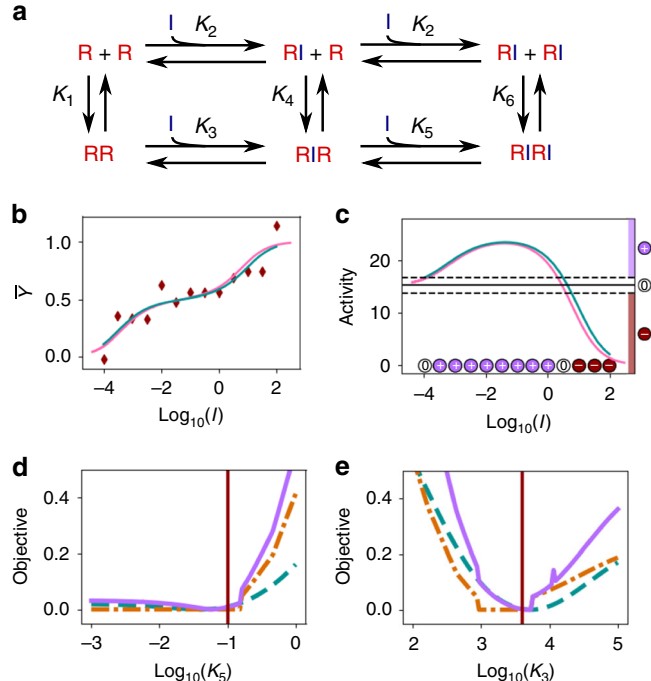

**Fig. 2** A model of Raf inhibition. **a** Reaction network. Raf (R) can dimerize, and each monomer can bind an inhibitor (I). **b** Synthetic quantitative data (dark red diamonds) of the free inhibitor concentration $I$ (in μM) versus the proportion of Raf bound to inhibitor $\bar{Y}$. **c** Synthetic qualitative data of free inhibitor concentration $I$ versus Raf activity. We assume Raf activity is proportional to RR + RIR, but the data (circles on $x$-axis) only tell us whether the activity is higher (+), lower (−), or within error (0) of the activity at $I = 0$. In **b**, **c** magenta curves indicate the ground truth, and blue curves indicate the best fit using the combined qualitative and quantitative datasets. Curves in **c** show the quantitative value of RR + RIR. **d**, **e** Profile likelihood uncertainty quantification of the two unknown parameters of the model, $K_3$ and $K_5$, using the quantitative data in **b** (blue dashed curve), the qualitative data in **c** (orange dash-dot curve) or both datasets combined (purple solid curve). Vertical red lines indicate the ground truth values. Vertical axes indicate the relative objective function values, obtained by normalizing each profile to zero by subtracting its minimum value

To test the capabilities of constrained optimization with this model, we generated synthetic datasets with the ground truth parameter values $K_1 = 0.04\ \mu M^{-1}$, $K_2 = 20\ \mu M^{-1}$, $K_3 = 4000\ \mu M^{-1}$, and $K_5 = 0.1\ \mu M^{-1}$. We assume a constant, known $R_{tot}$ concentration of 50 μM.

An experiment yielding quantitative data would be to measure a binding curve: measure the free inhibitor concentration $I$ versus receptor occupancy $\bar{Y}$ (Fig. 2b). Here we define $\bar{Y}$ as the fraction of Raf molecules bound to inhibitor, $\bar{Y} = (RI + RIR + 2RIRI)/R_{tot}$. A curve similar to this has been measured for EGF and the EGF receptor[19].

We consider a second experiment that yields only qualitative data, in which we measure free inhibitor concentration $I$ versus the overall Raf activity. We assume that the active Raf species are the uninhibited dimers RR and RIR.

We define $A = RR + RIR$ as a quantity proportional to Raf activity, but suppose that the experiment can only detect whether the activity is higher (+), lower (−), or within error (0) of the basal value of $A$ with no inhibitor (Fig. 2c). Note that a small concentration of inhibitor unintuitively causes an increase in Raf activity by increasing the concentration of the species RIR. This behavior is observed for many Raf inhibitors and poses a challenge for therapeutic applications[18].

We combine the quantitative and qualitative datasets by the method introduced in the previous section. Here, we are solving for the parameters $\mathbf{x} = (K_3, K_5)$.

For the quantitative term of the objective function $f_{quant}(\mathbf{x})$ (Eq. (2)), we directly use the synthetic data in Fig. 2b. For the qualitative term $f_{qual}(\mathbf{x})$ (Eq. (3)), we note that each data point in Fig. 2c can be expressed as an equality or inequality constraint on the value of $A$ at the particular value of $I$ given $\mathbf{x} = (K_3, K_5)$, which we will write as $A(I, \mathbf{x})$. Namely, for a "−" data point, we have $A(I, \mathbf{x}) < A(0)$, and an analogous constraint holds for a "+" point. For a "0" point, we have $|A(I, \mathbf{x}) - A(0)| < \epsilon$, where $\epsilon$ is the tolerance (we used $\epsilon = 1.5$ in this example). Note that $A(0)$, the value of $A$ with no inhibitor, is a known constant computed from $K_1$. Each constraint is then converted to the form $g_i(\mathbf{x}) < 0$ for use in Eq. (3) (e.g., for a "−" point at inhibitor concentration $I_i$, we have $A(I_i, \mathbf{x}) - A(0) < 0$).

We must choose the penalty constants $C_i$ in Eq. (3) based on our confidence in the qualitative data—a larger $C_i$ gives the qualitative data more weight compared to the quantitative data. We hand-selected a value of $C_i = 0.03$ for all $i$, to give roughly equal contributions from the qualitative and quantitative datasets (i.e., such that the blue and orange curves in Fig. 2d, e can be plotted using the same scale). Note that giving equal contributions to both datasets is an arbitrary choice for illustration—with real experimental data, the modeler may choose to give unequal weights based on the relative importance/credibility of the datasets.

We minimized $f_{tot}(\mathbf{x})$ by differential evolution, and found best-fit parameters of $K_3 = 5100\ \mu M^{-1}$ and $K_5 = 0.060\ \mu M^{-1}$, which are reasonably close to the ground truth.

To evaluate the strengths of combining quantitative and qualitative data, we performed uncertainty quantification. We used a variant of the profile likelihood approach, a method that is well-established for quantitative fitting[20]. One parameter of interest is held fixed, and the objective function is minimized by varying the remaining parameters. The resulting minimum is taken as the negative log likelihood of the fixed parameter value. The minimization is repeated for many possible fixed values of the parameter to produce a curve (Fig. 2d, e). Note that our objective function is not a likelihood in the rigorous sense, but has a similar interpretation in that a lower objective value indicates the parameter value is more consistent with the data.

We performed profile likelihood analysis using $f_{quant}(\mathbf{x})$, $f_{qual}(\mathbf{x})$, and $f_{tot}(\mathbf{x})$ in turn as the objective function, and considering two free parameters $K_3$ and $K_5$ (Fig. 2d, e). We found that each dataset individually provided bounds on possible parameter values, but the tightest bounds were obtained when we combined both datasets.

To make a numerical comparison, we can choose some maximum acceptable value for the relative objective function in Fig. 2d, e (we chose 0.15, for illustration), and find the range of possible parameter values consistent with that maximum objective. These ranges (Table 1) can be thought of as confidence intervals for each parameter, and we find the smallest intervals for the combined dataset. Notably, for $K_3$, each individual dataset provides a bound to within 2 orders of magnitude, whereas with

the combined datasets, we find a considerably tighter bound of 1.2 orders of magnitude.

Our results with this simple but biologically relevant model speak to the strengths of combining both qualitative and quantitative data for parameter identification.

**Fitting a model of yeast cell cycle control.** Having provided illustrations of the potential value of combining qualitative and quantitative data in two simple problems, we now apply the same method to identify parameters for a detailed model of yeast cell cycle control. We chose to analyze the model described in refs. [7,21], which has previously been used in an automated fitting effort. The previous fit[7] incorporated the phenotypes of 119 yeast strains (viable or inviable, but without considering the phase of cell cycle arrest). Nominal parameter values, based on earlier, hand-tuned models, were used to limit the search space of the optimization algorithm: The search covered a range of ±90% of each nominal value.

In a viable yeast cell, the expected model behavior is to have the volume $V$ of the cell increase over time, followed by cell division, which immediately reduces $V$. (Division of budding yeast is asymmetric. We arbitrarily choose to track the smaller daughter cell, and so multiply $V$ by 0.4 at division.) This cyclic increase in $V$ followed by division is expected to repeat in a stable, periodic manner. At the same time, the cyclins and other cell-cycle-dependent species tracked in the model are expected to oscillate periodically, each one peaking at a particular point in the cycle depending on its role. Finally, certain events must occur during each cycle: origin relicensing, bud formation, origin activation, and spindle assembly.

In addition to the previously used qualitative data on the viability of mutant yeast strains[6,7], quantitative time series data are available in the form of DNA microarray measurements of mRNA levels[22]. Although this dataset has a high level of noise, it contains useful, additional information such as at which point in the cell cycle each species reaches its peak value.

As with the Raf example, we considered the objective function $f_{tot}(\mathbf{x})$ as described in Eq. (1). Here, $\mathbf{x}$ is a vector of 153 model parameter values.

In this model, the constraints relate to properties of entire time courses, rather than individual data points. Some examples are shown in Fig. 3. The constraints include those that require that a variable never reaches a particular value (row i), or must reach a value at some point in the simulation (row ii). It is also possible for a constraint to be enforced over only a particular window of times (row iii). As shown with the examples in Fig. 3, each constraint can still take the form $g_i(\mathbf{x}) < 0$, and $f_{qual}(\mathbf{x})$ takes the same form as in Eq. (3).

For each possible phenotype that exists in the data (viable, G1 arrest, etc.), we generated as many constraints as possible that enforce known properties of that phenotype (Supplementary Table 1). To model each mutant yeast strain, we made perturbations to appropriate model parameters, as summarized in Supplementary Table 2. We ran one simulation per mutant strain, and enforced all constraints corresponding to the mutant's phenotype. The 119 strains summarized in Supplementary Table 2 resulted in a total of 1647 constraints included in the Eq. (3) summation.

In this example, we used different static penalties $C_i$ for different constraints. We chose the $C_i$ to account for how the degree of constraint violation can differ by orders of magnitude, depending on the constraint. For example, in Fig. 3, row ii, the maximum possible constraint violation is 1 (if SPN remains at 0), whereas in Fig. 3, row iii, the violation can be over 100 if ORI starts increasing exponentially at time 0. The $C_i$ were hand-chosen to offset these differences in magnitude, such that each

**Table 1 Bounds on parameters (in $\mu M^{-1}$) using quantitative, qualitative, or both datasets, assuming a maximum relative objective function value of 0.15**

| Parameter | Ground truth value | Quant. | Qual. | Combined |
|---|---|---|---|---|
| $\log_{10}(K_3)$ | 3.6 | 2.8–4.8 | 2.6–4.6 | 3.0–4.2 |
| $\log_{10}(K_5)$ | −1.0 | <−0.1 | <−0.5 | <−0.6 |

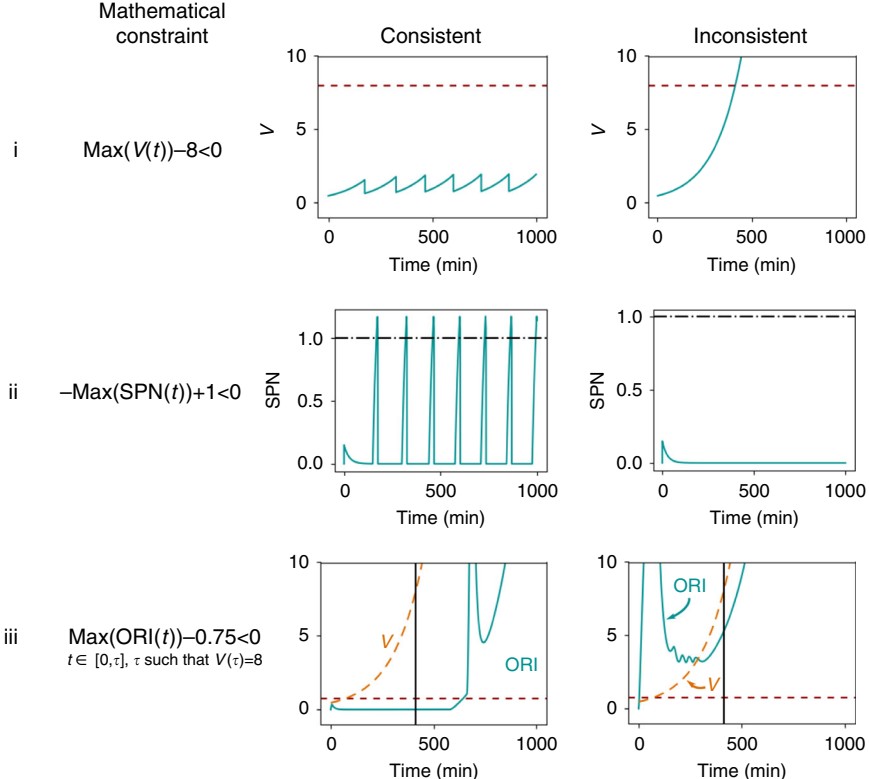

**Fig. 3** Example constraints used in parameterizing the yeast cell cycle model. The constraints are based on known properties of particular cell phenotypes: Row (i) In a viable cell, the cell must divide before the normalized volume $V$ reaches 8. Row (ii) In a viable cell, the assembly of the mitotic spindle SPN must finish (reach 1). Row (iii) In an inviable cell in G1 arrest, origin activation ORI does not finish (fails to reach 1). Each property was converted to an inequality constraint on one of the outputs of the model. Simulations of cells corresponding to the particular phenotype should be consistent with the constraint (left panels). Simulations of cells corresponding to some other phenotype may not be consistent with the constraint (right panels). Red broken lines indicate that the trace must never exceed the indicated value, and black dash-dot lines indicate the trace must exceed the value at some point. In row iii, the constraint is only enforced to the left of the vertical line, which is located at time $\tau$ such that $V(\tau) = 8$

constraint contributed roughly equally (values are given in Supplementary Table 1). Additionally, we note that although most constraints should be treated as soft constraints as in the Raf example, we can be absolutely certain about the constraints on wild type cells. Therefore, for constraints on wild type cells, we used $C_i$ 1000 times larger than the $C_i$ for corresponding constraints on mutant cells.

$f_{quant}(\mathbf{x})$ is once again a standard sum of squares error function based on DNA microarray time courses for 10 species monitored under three experimental conditions[22] (Fig. 4d, f and Supplementary Fig. 1).

When constructing $f_{tot}(\mathbf{x})$ (Eq. (1)), we scaled the $C_i$ in Supplementary Table 1 by a factor of 1/15. This factor was tuned by trial and error such that the best fit achieved good agreement with both the qualitative and quantitative data (as opposed to focusing on only one of the datasets). We minimized $f_{tot}(\mathbf{x})$ using a version of enhanced scatter search, an evolutionary optimization algorithm designed for high-dimensional problems[15,23,24]. Our optimization algorithm considered a search space spanning ±2 orders of magnitude around the nominal values of ref. [7].

Our best-fit parameter set (Supplementary Table 3) achieved good agreement with both the quantitative and qualitative datasets. Consistent with the qualitative data, outputs of the model have different behaviors depending on the known phenotype of the mutant (Fig. 4a–c and Supplementary Fig. 2). For example, the output ORI has an oscillatory time course in viable cells, remains near 0 in a mutant with a G1 arrest phenotype, and increases without bound in a mutant with a telophase arrest phenotype.

Our best fit also shows good agreement with the quantitative data (Fig. 4d–f and Supplementary Fig. 1). Note that our fit matches the cell cycle length that is suggested by the experimental time series, and also has the level of each species reach a peak at a point in the cell cycle consistent with these data.

Our fit achieved objective function values $f_{quant}(\mathbf{x}) = 42$, and $f_{qual}(\mathbf{x}) = 420$. The performance of our parameter sets for each individual constraint is shown in Supplementary Table 2. For comparison, we also show the performance of the fit reported by Oguz et al. (ref. [7]).

We also want to compare our overall quality of fit to that achieved in ref. [7]. For the quantitative dataset, our fit outperforms that of ref. [7], which has an objective function value of $f_{quant}(\mathbf{x}) = 81$. This result is unsurprising given that ref. [7] did not include the quantitative data in the optimization. In fact, it is notable that the fit of ref. [7] performs so well on the quantitative data (within 2-fold of our result), which could be seen as an independent confirmation of that fit. To compare performance on the qualitative data, it is not fair to directly compare $f_{qual}(\mathbf{x})$—in some cases where both parameter sets are inconsistent with a particular qualitative observation, the fit of ref. [7] incurred a very high penalty, whereas our fit optimized $f_{qual}(\mathbf{x})$ to lower the penalty. Although the penalty functions are useful in guiding our fit to a good result, they are not as useful for comparison against a fit obtained by another method. We instead use another evaluation metric, used in ref. [7], which is to count the number yeast strains (out of the total 119) for which the model predicts a phenotype consistent with the data (i.e., satisfies all system properties in Supplementary Table 1). We score the model

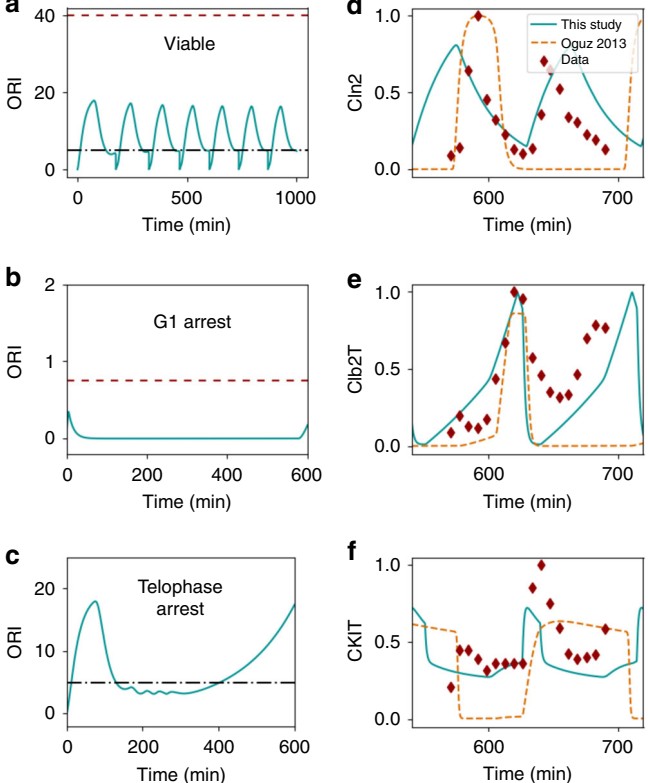

**Fig. 4** The yeast cell cycle model is fit to both qualitative and quantitative data. **a–c** Three different yeast strains satisfy different requirements with respect to the output variable ORI. Black dash-dot lines indicate a value that the time course must reach at some point, and red broken lines indicate a value the time course must never reach. **a** Wild type, viable cell. **b** *cln3Δ bck2Δ* mutant, with a G1 arrest phenotype **c** *cdc14-ts* mutant, with a telophase arrest phenotype. **d–f** Fits to quantitative data (red diamonds) for three selected model output variables. Our fits (solid blue curves) are compared to the results from ref. [7] (orange broken curves), which were not fit to quantitative data

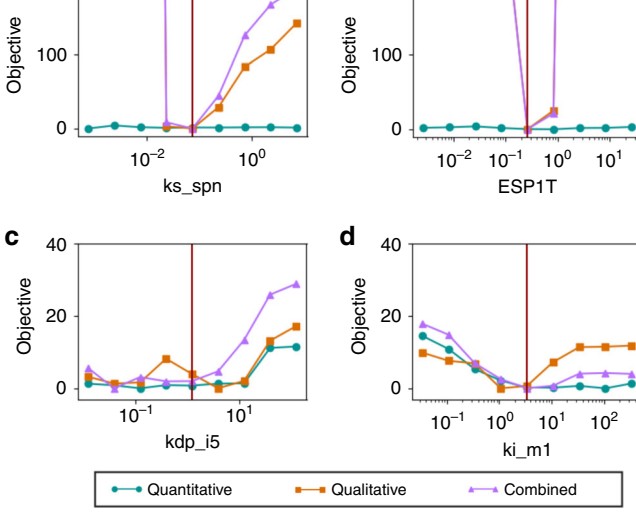

**Fig. 5** Profile likelihood uncertainty quantification. We calculated profile likelihood for selected parameters in the yeast model: **a** ks_spn, **b** ESP1T, **c** kdp_i5, **d** ki_m1. For each parameter, we compare the profiles generated from only quantitative data (blue circles), only qualitative data (orange squares), or both datasets combined (purple triangles). Sampling of the profiles was coarse due to the computational cost of fitting. At each point shown, we performed five replicates of the optimization and took the best result. Each profile was normalized to zero by subtracting the minimum value. The vertical, red lines indicate our best-fit value for each parameter

as consistent with a strain if the total penalty value for that strain is below a threshold (either zero or a small positive value). This evaluation metric effectively checks that each constraint of interest is satisfied, without overpenalizing large constraint violations. At any chosen threshold, our fit outperforms the fit of ref. [7] in terms of number of consistent strains (Supplementary Fig. 3).

As with the Raf model, we assessed the uncertainty of model parameters using a profile likelihood approach (Fig. 5). We computed profiles using $f_{quant}(\mathbf{x})$, $f_{qual}(\mathbf{x})$, and $f_{tot}(\mathbf{x})$. Again we see that by using both the quantitative and qualitative datasets, we achieve the tightest bounds on the model parameters. For several parameters (Fig. 5a, c, d, left side of panel), the profile for the combined datasets had higher relative objective values than the profiles for either individual dataset, suggesting a contribution from both datasets. Other parameters (Fig. 5b, d, right side of panel) were constrained primarily by the qualitative data.

The profile likelihood results demonstrate the value of combining quantitative and qualitative data, in the context of this detailed model of yeast cell cycle control which is the subject of current active research.

## Discussion

We have demonstrated how qualitative and quantitative data may be used together for parameter identification in biological modeling. We show the generality of the approach with three examples

of increasing complexity: a toy problem with polynomial functions, a biologically relevant but simple model of Raf inhibition, and a detailed model of cell cycle control in yeast. In each case, we were able to apply penalty function-based constrained optimization because there were qualitative data available, which could be expressed as inequality constraints on the outputs of the model.

We used one particular optimization algorithm (a variant of scatter search), which yielded an acceptable fit for our model of interest. It remains an avenue for future work to determine which algorithms are most effective in the general case for constrained optimization in systems biology.

We enforced qualitative data by means of static penalty functions. A known limitation of this method is the need to choose penalty constants $C_i$ for each constraint[25]. Although progress has been made in developing constrained optimization algorithms that do not require hand-chosen constants[8], future work is required to adapt such algorithms to biological applications—in particular, available algorithms are not designed for problems such as that of the yeast cell cycle model, in which not every constraint is satisfied in the best fit. At the end of this section, we discuss how to handle the limitations of static penalty functions by making reasonable choices for the $C_i$.

Our work addresses what has been a fundamental disconnect between experimentalists and modelers. Modelers want quantitative time courses to identify model parameters, but experimentalists are reluctant to acquire time courses, which are more costly than qualitative or semi-quantitative measurements. Our approach allows modelers to make use of data that experimentalists may more easily generate, and data that already exists in the literature. Every successful experiment provides at least a qualitative statement about the biological system of interest, and many such statements can be cast as inequality constraints. As we have shown, these constraints can be used in fitting to improve confidence limits on model parameters.

We expect that our approach will open new opportunities for the development of experimentally validated models for cellular networks. We now provide some practical advice for modelers seeking to apply the methodology presented here to new biological questions.

To apply our methodology, one must first obtain qualitative data pertaining to the system of interest. Examples of assembled qualitative datasets can be found in refs. [6,26]. Next, one must define the objective function $f_{tot}$ (Eq. (1)). A user-defined setting in the objective function is the weight $C_i$ of each constraint. In the absence of additional information, a good choice is to use the $C_i$ to normalize to the order of magnitude of the model output. For example, if one constrained output is of order 1 and another is of order 100, the $C_i$ for the latter should be smaller by a factor of 100. If additional information is available indicating that some constraints are more strict than others, this information can be incorporated by increasing the appropriate $C_i$.

The $C_i$ affect the relative weighting of the qualitative and quantitative data, and the modeler must choose the relative importance. To help choose $C_i$, we suggest performing optimization on $f_{quant}$ alone, and then on $f_{qual}$ alone. Then, all the $C_i$ can be scaled such that the optimal values of $f_{quant}$ and $f_{qual}$ are in the desired proportion (this is not identical to directly optimizing $f_{tot}$, but is a useful heuristic to set $C_i$).

Finally, one must choose an optimization algorithm to find the best-fit parameter set. For biological models with a large number of parameters, metaheuristic algorithms such as differential evolution or scatter search are likely to be effective. Software packages implementing these algorithms are freely available[27,28]. The software BioNetFit[29] is under active development by the authors of the present study, and in the near future will streamline constrained optimization for models defined in SBML[30] or BioNetGen language (BNGL)[31].

## Methods

**Polynomial model**. A system of linear equations for computing the polynomial coefficients was implemented with the Python Numpy module. We suppose we have $n + 1$ equally spaced qualitative data points, which divide the range $x = [0, 10]$ into $n$ equally-sized intervals. Exactly two of these intervals will be bounded on one end by a "−" point and on the other by a "+" point; these two intervals must contain the intersection points $x_1$ and $x_2$.

To generate Fig. 1b, within the possible intervals for $x_1$ and $x_2$, we sampled all possible combinations of 100 (evenly spaced) values of $x_1$ and 100 values of $x_2$, and solved for the coefficients. Considering only the fits that yielded positive values for all coefficients, we report the minimum and maximum possible values for each coefficient.

**Raf inhibitor model**. The model shown in Fig. 2a was implemented in Python (Supplementary Software 1). We generated synthetic data points at various values of $I$ and constant $R_{tot} = 50 \, \mu M$, and populated the other species concentrations based on the equations of the model. To generate quantitative data, we calculated $\bar{Y} = (RI + RIR + 2RIRI)/R_{tot}$, and perturbed results by adding normally distributed noise with standard deviation 0.1. To generate qualitative data, we calculated $A = RR + RIR$, and compared the result to the value $A(0) = 15.24$. If the difference from $A(0)$ was <1.5, the data point was labeled as "0" (within error), otherwise it was labeled as "+" or "−" appropriately.

For fitting, the objective function was created as described in Results. Minimization was performed with the Scipy function optimize. differential_evolution(), with a maximum of 1000 iterations, strategy of "best1exp", and search range of $[10^{-4}, 10^4]$ for each parameter.

To perform profile likelihood analysis, for each parameter, we considered 100 possible fixed values (log uniformly distributed in the range $[10^2, 10^5]$ for $K_3$ and $[10^{-3}, 10^0]$ for $K_5$). At each fixed value considered, minimization was performed by the same method as above, except that the parameter of interest was held at the fixed value. We report the resulting minimum objective function values in the profile likelihood plots (Fig. 2d, e).

**Yeast cell cycle model**. The yeast model was implemented in C++ code adapted from ref. [7]. We edited this code to be callable from Python, and to include additional output variables required for our constraints.

Fitting was performed using purpose-built Python code (Supplementary Software 1). To specify qualitative constraints, we generated a text file that maps parameter changes for a yeast mutant to a list of constraints to be applied to that mutant. Our code reads the text file to generate a qualitative objective function. To use quantitative data in fitting, we first obtained the raw DNA microarray data from Spellman et al.[22]. We used the three datasets that were fully analyzed in the original study: cells synchronized with alpha factor, a *cdc15-ts* mutation, or a *cdc28-ts* mutation (a dataset originally from Cho et al.[32]). From each dataset we extracted the mRNA levels corresponding to all proteins included in the cell cycle model. The data were renormalized such that the highest point in each time course had a value of 1, and these points were used as our input to generate a quantitative objective function. Since we do not know a priori where within in our model's cycle the synchronized cells started from, we added one additional model parameter for each of the 3 experimental conditions: $\phi_\alpha$, $\phi_{cdc15}$, $\phi_{cdc28}$, representing the time offset at which we should start fitting to the quantitative data. To make a fair comparison with the fit of ref. [7] (Fig. 4d–f and Supplementary Fig. 1), we also selected $\phi$ values for that parameter set that led to its best possible fit.

To compute the overall objective function for a given parameter set, 123 simulations were run. For each mutant strain in the data (119 total), we ran one simulation and totaled any penalties associated with the constraints for that strain's phenotype. For the quantitative data, one simulation with wild type parameters was run for each of the three datasets, and the sum-of-squares error was calculated for each simulation output. In these simulations, the parameter for mean doubling time (MDT) was set to the average value suggested in ref. [22]: 66 for alpha factor, 70 for *cdc15-ts*, and 90 for *cdc28-ts*.

Finally, it was necessary to add two problem-specific terms to the objective function. For wild type cells, we ran a separate simulation at MDT = 90, and required that the values of volume $V$ at each of the last three cell divisions of the simulation were almost equal to each other. A penalty term proportional to the difference in $V$ was added. This prevents fits that alternate between two or more sizes at division, which would be non-biological behavior. A similar restriction was used in ref. [7]. Second, we observed the possibility of time courses that rapidly oscillate to overfit the quantitative data. To avoid these fits, we imposed a penalty for each time course of a quantitative variable that contained more than 20 local maxima. Because each simulation includes only ≈7 cell cycles, a reasonable time course should have <20 local maxima.

The objective function was minimized using scatter search, an evolutionary optimization algorithm[15]. We incorporated some features from enhanced scatter search[23,24]. Our specific implementation of scatter search is summarized as follows. A reference set was maintained consisting of 12 parameter sets. In each iteration, a parameter set was proposed for each possible pair of elements in the reference set ($12 \times 11$ total per iteration), by the combination method described by Egea et al.[23]. If a parameter went outside the search range (prior value in the study of Oguz et al.[7] ± two orders of magnitude) it was reset to the boundary of the range. A proposed parameter set replaced its own parent in the reference set if it had a lower objective value. If a parameter set remained for five iterations without improvement, it was assumed to be a local minimum, and was replaced with a new random parameter set. To avoid re-sampling the same area of parameter space, the new random parameter sets were drawn from a queue that was initialized with 70,000 Latin-hypercube-distributed random parameter sets at the start of fitting. The fitting was run for 70,000 iterations, totaling $1.1 \times 10^9$ individual simulations.

Profile likelihood analysis was performed as follows. At each point in the profile likelihood plot, the indicated parameter value was held fixed. Minimization of the appropriate objective function ($f_{quant}$, $f_{qual}$, or $f_{tot}$) over the remaining parameters was performed by differential evolution (chosen here instead of enhanced scatter search because it provided faster convergence in the smaller search space used in these calculations), with each parameter allowed to vary ±50% from its best fit value. The population size was 80. Optimization was run until the population converged, defined as less than a 1% difference in the objective values between the best and worst members of the population, up to 2000 iterations. Five replicates were performed for each point.

**Code availability**. Python and C++ source code for the Raf inhibitor and yeast cell cycle models, as well as implementations of both models in SBML format, are provided as Supplementary Software 1.

## Data availability

All data generated or analyzed during this study are included in this published article and its supplementary information files.

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

## Acknowledgements

This work was supported by a grant (R01GM111510) from the National Institute of General Medical Sciences of the National Institutes of Health and the Laboratory Directed Research and Development program at Los Alamos National Laboratory, which is operated for the National Nuclear Security Administration of the U.S. Department of Energy under contract DE-AC52-06NA25396. We thank Fred Glover, Gary Kochenberger, and Nick Hengartner for helpful discussions.

## Author contributions

W.S.H. and R.G.P. designed the study. E.D.M. and R.D. wrote computer code, performed simulations, and analyzed the results. E.D.M. and W.S.H. wrote the manuscript. All authors read and approved the final manuscript.

## Additional information

**Competing interests:** The authors declare no competing interests.

