## [Peer Review File · Nature Communications]

Reviewers' comments:

Reviewer #1 (Remarks to the Author):

Review for Nature Communications

By John Tyson and Pavel Kraikivsky, Virginia Tech

The paper by Mitra, Hlavacek and others is a useful contribution to an important and difficult problem, namely "parameter identification for systems biology models" that must be judged in light of experimental data including both qualitative observations and quantitative measurements. As in any parameter optimization problem, there are two aspects to the solution: first, to define an objective function that melds the quantitative and qualitative observations into a single real number, the objective function $f_{\text{tot}}(\mathbf{x})$, that is to be minimized with respect to variations in the parameter vector \mathbf{x} ; and second, to propose an optimization scheme (local or global, deterministic or stochastic, or some hybrid approach) that searches for a minimum value of f_{tot} at some point \mathbf{x}_{min} in parameter space.

To illustrate the potential value of their approach to this problem, Mitra et al. first present two simple examples: a mathematical illustration of fitting curves to data points subject to inequality constraints, and a model of Raf-kinase inhibition, optimized against "synthetic data" derived from the model itself. Then they buckle down to a more realistic problem: to identify parameters in a model of cell cycle control in budding yeast on the basis of both qualitative observations (the phenotypes of mutant yeast strains) and quantitative measurements (temporal fluctuations in gene expression in WT cells). The model, due to Oguz et al. (2013), contains 153 parameters; the qualitative data set consists of the phenotypic characteristics of 119 mutant strains of yeast; and the quantitative data set consists of time courses of mRNA expression from 10 cell-cycle genes in WT cells under three different synchronization protocols.

The authors carefully describe how they define the objective function and briefly describe the minimization algorithm that they employed (enhanced scatter search). They do not provide the C++ code used to simulate the model or the Python code used to calculate and minimize the objective function. Their results are provided in Suppl Table S3, which gives the "initial value" and "final fit value" of each of the 153 parameters in the model. They do not say where they obtained the "initial" parameter values; presumably they are "reasonable" first guesses based on parameter ranges given in Oguz et al. Curiously, the Oguz paper never reports the "best-fitting" parameter values obtained by their parameter-fitting approach.

The authors claim that the performance of their optimization procedure is considerably better than the optimization procedure used by Oguz: 2-fold better in fitting the quantitative data, and 13.6-fold better in fitting the qualitative data. It is not surprising that Mitra does better than Oguz in fitting the quantitative data, because Oguz et al. did not try to fit their model to the gene expression data. What is surprising is that the Oguz model does as well as reported here in fitting the gene expression data, which is unrelated to the mutant phenotypes. One might consider these results of Mitra et al. as independent confirmation of the Oguz model and parameter fit.

We were surprised that the Oguz model is reported to be so awful in fitting the "mutant phenotype" data ($f_{\text{qual}} = 5711$ for Oguz, versus $f_{\text{qual}} = 420$ for Mitra). Of the 119 mutant strains in the qualitative data set, the Oguz model/param set successfully accounted for the phenotype (viable or inviable) of ~110

strains. The Mitra param set does not do any better than this (probably a little worse in terms of total # of strains correctly predicted as viable or inviable). The large difference in f_{qual} scores is due to the objective function used by Mitra to enforce additional qualitative constraints that require viable mutants to execute certain cell cycle events in the correct order, and inviable mutants to arrest at the observed phase of the cell cycle (G1, S/G2, metaphase, etc.). We are completely in agreement with Mitra et al. that these additional constraints should be enforced, but there seems to be a serious problem in how one of these constraints is implemented.

Looking closely at Table S2, one sees that all of the large penalties (penalties > 100) incurred by the failed constraints in the Oguz model/param set are attributable to a single constraint, that $\max(\text{ORI}(t)) < \text{"threshold"}$, where "threshold" is 0.75 for inviable, G1-arrested strains and 40 for viable strains. The sum total of all these ORI-penalties is ~4650, or 80% of the $f_{\text{qual}} = 5711$ score for the Oguz model. If we discount the ORI-penalties, then the Oguz score ($f_{\text{qual}} = 1060$) is only about 2-fold larger than the Mitra score, which is unsurprising since Oguz did not even try to correctly simulate these additional constraints.

Why should we discount the ORI-penalties? Mitra et al. have penalized the parameter fit if the $\text{ORI}(t)$ variable gets too large compared to a "threshold" of their choosing. But I think this constraint comes from a misunderstanding by Mitra et al. of the meaning of $\text{ORI}(t)$. $\text{ORI}(t)$ is a "flag" in the Oguz model. It is reset to 0 in a newborn cell, and its function is to integrate the temporally changing activities of the Clb cyclins during G1 phase of the cell cycle, because Clb5- and Clb2-dependent kinases are primarily responsible for phosphorylating origins of replication in the yeast cell genome and thereby triggering the onset of S phase. As long as $\text{ORI}(t) < 1$, the cell is in G1 phase; as soon as $\text{ORI}(t)$ exceeds 1, the cell enters S phase. After that decision point is past, the model takes no further notice of the value of $\text{ORI}(t)$. It can increase to any arbitrary value, because it is reset to 0 when the cell divides. So no penalty should be given for $\text{ORI}(t)$ getting larger than 40 in a viable strain. Of course, a G1-arrested cell should be penalized if $\text{ORI}(t)$ exceeds 1.0 (or 0.75, we won't quibble), but the penalty should not be outrageously high as in Table S2.

We think this problem needs to be corrected and the best-fits need to be recalculated for both optimization methods. As the procedure currently stands, the objective function overweights the ORI variable, which will force the optimization routine to choose parameter values that limit the growth of $\text{ORI}(t)$ after it crosses $\text{ORI} = 1$, even though the subsequent increase in ORI is inconsequential to the model.

Similar considerations apply to $\text{BUD}(t)$ and $\text{SPN}(t)$, which are also flags to determine the timing of bud emergence and spindle assembly. A few minor problems should also be addressed:

- (1) In viable cells, $[\text{Clb2}]$ must drop below K_{EZ} as cells exit mitosis, in order to relicense origins of replication. We don't see that this constraint is included in Mitra's objective function.
- (2) What is the meaning of the "FLAG" variables ($\text{FLAG-BUD}(t)$, $\text{FLAG-UDNA}(t)$, etc.) in Table S1?
- (3) It would help many readers if the authors were to include a brief description of "enhanced

scatter search" in the section on Materials and Methods.

In summary, we think that Mitra et al. have clearly described a new and valuable approach to parameter estimation in large, realistic, mathematical models of molecular regulatory mechanisms in systems biology. They have described a reasonable way to incorporate both qualitative and quantitative data into a single, scalar objective function, and they have explored the feasibility of a reasonable method (enhanced scatter search) to minimize the objective function. But the details of their calculations are seriously marred by a misunderstanding of the roles of the "flags" in the Oguz model. By overweighting the importance of these flags, they may have skewed their optimization routine into regions of parameter space that are not particularly relevant to the spirit of the mathematical model. This mistake seriously compromises the conclusions they draw from their calculations, especially the surprisingly poor performance of the Oguz model on the mutant-phenotype observations. We recommend that the objective function be modified to properly account for the "flags" and that the calculations be repeated with the new objective function.

Reviewer #2 (Remarks to the Author):

Mitra et al. present a case study on using qualitative data for calibrating quantitative dynamical models. The manuscript also progresses toward a formalism for using qualitative data to calibrate models and provides practical advice on how to use qualitative data to calibrate models, including how to choose weights in error functions and algorithms for constrained optimization. The manuscript is a valuable contribution because it highlights an abundant source of data (qualitative data) that could, but is often not, used to calibrate models. This is particularly important for building larger models that represent each gene product that are needed to support medicine and bioengineering. The manuscript is well-written and well-organized. However, we feel that the manuscript could have more impact on the field with a few small improvements to the text.

Major comments

The manuscript could be strengthened by adding a brief section which presents a formal definition of the model calibration using quantitative and qualitative data. The manuscript does present a definition in Section 2.2.2. Nevertheless, it would be helpful for readers to present this separately from the individual case studies.

The manuscript could be strengthened by adding a brief section with practical advice on how to use both quantitative and qualitative data for calibration. This could include recommendations for databases to obtain qualitative data, constrained optimization algorithms, optimization packages. The manuscript does have recommendations throughout the text. Nevertheless, it would be helpful to organize these recommendations into a single section.

The manuscript could be strengthened by adding a brief exposition on the similarity of the approach presented to model checking/validation.

Minor comments

Section 2.1 could be eliminated for brevity.

Please clarify the qualitative error function $g_i(x)$ on page 4. Is this Boolean-valued or $\max\{0, A(I_i, x) - A(0)\}$?

It would be interesting to comment on why this error function, $g_i(x)$ was chosen instead of alternatives, such as the difference ($\max\{0, A(I_i, x) - A(0)\}$) or squared difference ($\max\{0, A(I_i, x) - A(0)\}^2$).

The second example is informative, but the evidence in Section 2.2.3 and Table 1 that using both quantitative and qualitative data outperforms using only quantitative or qualitative data is weak. Other examples may be able to more clearly demonstrate the utility.

The comparison between using both quantitative and qualitative data to using only quantitative data in Section 2.3.2 lacks context. The demonstration would be bolstered by independently validating the revised parameter values with new experimental data.

In Section 2.3, where the yeast cell cycle model is introduced, please clarify what data set/types were used for parameter identification in the original study. Currently, this information is not clear. This makes it harder to evaluate the comparison of model fits (Figure 4).

Small legends would improve the readability of Figures 4 and 5 by enabling the reader to focus on the figure rather than reading the caption.

Reviewer #3 (Remarks to the Author):

Combining qualitative and quantitative data for parameter identification is a very valuable goal. The authors aim to reach this goal by adding a term to the standard (quantitative data) objective function that penalizes a violation of the qualitative data. They exemplify their approach on two simple simulated data sets and on real world data from the yeast cell cycle control.

While the proposed methods is intuitive, my main criticism is that it lacks any statistical rigour. It starts with eq. (2) in which each term of the sum should be divided by the variance of the measurement error to turn it into a maximum likelihood estimator that reflects the confidence in the quantitative data. As the authors state correctly, C in eq. (3) reflects the confidence in the qualitative data. But the next sentence says: 'We hand-selected a value of $C=0.03$ to give roughly equal contributions from the qualitative and qualitative data sets'. There is no reason why these contributions should be equal. The situation becomes worse if there are many C_i . Section 2.3.1: 'The C_i were hand-chosen ... such that each constraint contributed roughly equally.' Why should they? One constraint might be very strict, another rather loose. Last paragraph of Sec. 2.3.1 'This factor [C_i scaling] was tuned by trial and error ...'

In conclusion, the proposed methods replaces the hand-tuning of parameters by Novak & Tyson to come up with there first cell cycle model by hand-tuning of C_i s.

We appreciate the constructive feedback from the reviewers, and have made revisions in response to the reviewer comments as described in the detailed points below.

Changes we have made to the main text are highlighted in red. Changes to the Supplementary Information are not highlighted, but are described below in the detailed responses. We have added a total of 14 new references: reference numbers 2, 4, 5, 9-11, 14, and 25-31.

We hope that the manuscript is now suitable for publication in *Nature Communications*.

Reviewer #1 (Remarks to the Author):

By John Tyson and Pavel Kraikivsky, Virginia Tech

The paper by Mitra, Hlavacek and others is a useful contribution to an important and difficult problem, namely "parameter identification for systems biology models" that must be judged in light of experimental data including both qualitative observations and quantitative measurements.

Response 1.1: We are delighted that Drs. Tyson and Kraikivsky found our manuscript to be a useful contribution that addresses an important problem.

As in any parameter optimization problem, there are two aspects to the solution: first, to define an objective function that melds the quantitative and qualitative observations into a single real number, the objective function $f_{\text{tot}}(x)$, that is to be minimized with respect to variations in the parameter vector x ; and second, to propose an optimization scheme (local or global, deterministic or stochastic, or some hybrid approach) that searches for a minimum value of f_{tot} at some point x_{min} in parameter space.

To illustrate the potential value of their approach to this problem, Mitra et al. first present two simple examples: a mathematical illustration of fitting curves to data points subject to inequality constraints, and a model of Raf-kinase inhibition, optimized against "synthetic data" derived from the model itself. Then they buckle down to a more realistic problem: to identify parameters in a model of cell cycle control in budding yeast on the basis of both qualitative observations (the phenotypes of mutant yeast strains) and quantitative measurements (temporal fluctuations in gene expression in WT cells). The model, due to Oguz et al. (2013), contains 153 parameters; the qualitative data set consists of the phenotypic characteristics of 119 mutant strains of yeast; and the quantitative data set consists of time courses of mRNA expression from 10 cell-cycle genes in WT cells under three different synchronization protocols.

The authors carefully describe how they define the objective function and briefly describe the minimization algorithm that they employed (enhanced scatter search). They do not provide the C++ code used to simulate the model or the Python code used to calculate and minimize the objective function.

Response 1.2: We intended to provide Supplementary Data 1, a ZIP file containing the Python and C++ code for fitting the yeast cell cycle model (folder YeastPython), an SBML implementation of the yeast cell cycle model (Yeast.xml), a Python implementation of the Raf model (RAFmodel.py), and an SBML implementation of the Raf model (Raf.xml). We are unsure why Drs. Tyson and Kraikivsky were not able to access this ZIP file; it may have been a file upload error of ours. We have tried our best

to make sure the ZIP file has been uploaded correctly. We hope that the reviewers are able to access the file in our revised submission.

Their results are provided in Suppl Table S3, which gives the "initial value" and "final fit value" of each of the 153 parameters in the model. They do not say where they obtained the "initial" parameter values; presumably they are "reasonable" first guesses based on parameter ranges given in Oguz et al. Curiously, the Oguz paper never reports the "best-fitting" parameter values obtained by their parameter-fitting approach.

Response 1.3:

For clarity, we have changed the column label in Table S3 from "Initial value" to "Center of log uniform search space" We have updated the legend to explain the meaning of this column:

"In the study of Oguz et al. 2013 (ref. 7), the search space was chosen based on a nominal estimate of each parameter value, given in column 2 below. Their fitting algorithm considered a uniform search space spanning up to $\pm 90\%$ of these nominal values. For example, for the parameter *gamma* with an estimated value of 1, their search space ranged from 0.1 to 1.9. We took a similar approach, and centered our search space on the same nominal values as ref. 7, but considered a larger search space: A log uniform space spanning ± 2 orders of magnitude from the nominal values. For example, for the parameter *gamma*, we considered values ranging from 0.01 to 100. Our methodology would be expected to find a good fit even if the nominal values were inaccurate (by up to 2 orders of magnitude)."

The authors claim that the performance of their optimization procedure is considerably better than the optimization procedure used by Oguz: 2-fold better in fitting the quantitative data, and 13.6-fold better in fitting the qualitative data. It is not surprising that Mitra does better than Oguz in fitting the quantitative data, because Oguz et al. did not try to fit their model to the gene expression data. What is surprising is that the Oguz model does as well as reported here in fitting the gene expression data, which is unrelated to the mutant phenotypes. One might consider these results of Mitra et al. as independent confirmation of the Oguz model and parameter fit.

Response 1.4: We have added the point in Section 2.4.2 that the good performance of the fit of Oguz et al. (2013) on the quantitative data could be seen as an independent confirmation of the fit.

We were surprised that the Oguz model is reported to be so awful in fitting the "mutant phenotype" data (f qual = 5711 for Oguz, versus f qual = 420 for Mitra). Of the 119 mutant strains in the qualitative data set, the Oguz model/param set successfully accounted for the phenotype (viable or inviable) of ~110 strains. The Mitra param set does not do any better than this (probably a little worse in terms of total # of strains correctly predicted as viable or inviable). The large difference in f qual scores is due to the objective function used by Mitra to enforce additional qualitative constraints that require viable mutants to execute certain cell cycle events in the correct order, and inviable mutants to arrest at the observed phase of the cell cycle (G1, S/G2, metaphase, etc.). We are completely in agreement with Mitra et al. that these additional constraints should be enforced, but there seems to be a serious problem in how one of these constraints is implemented.

Looking closely at Table S2, one sees that all of the large penalties (penalties > 100) incurred by the failed constraints in the Oguz model/param set are attributable to a single constraint, that $\max(\text{ORI}(t)) < \text{"threshold"}$, where

"threshold" is 0.75 for inviable, G1-arrested strains and 40 for viable strains. The sum total of all these ORI-penalties is ~4650, or 80% of the f qual = 5711 score for the Oguz model. If we discount the ORI-penalties, then the Oguz score (f qual = 1060) is only about 2-fold larger than the Mitra score, which is unsurprising since Oguz did not even try to correctly simulate these additional constraints.

Why should we discount the ORI-penalties? Mitra et al. have penalized the parameter fit if the ORI(t) variable gets too large compared to a "threshold" of their choosing. But I think this constraint comes from a misunderstanding by Mitra et al. of the meaning of ORI(t).

Response 1.5: Drs. Tyson and Kraikivsky are correct to point out that in the context of the model, ORI(t), BUD(t), and SPN(t) only have meaning in terms of when they cross values of 1, and so in principle, it is acceptable for them to increase to an arbitrarily large value.

However, it is required for these variables to have oscillatory behavior in viable cells, with a local maximum in each cell cycle. As a surrogate for "has oscillatory behavior", which can be difficult to express in terms of inequality constraints, we chose to impose upper bounds on each of the variables - this proved effective because the alternative to oscillatory behavior in this model is for the variable to increase without bound. The particular values chosen for these upper bounds are not meaningful - we chose them during our initial exploration of parameter space with the goal to choose values that in practice, oscillatory time courses would not cross, but monotonically increasing time courses would cross. We have made this motivation more clear in the descriptions of these constraints in Table S1, and in a new footnote to Table S1 associated with these constraints.

ORI(t) is a "flag" in the Oguz model. It is reset to 0 in a newborn cell, and its function is to integrate the temporally changing activities of the Clb cyclins during G1 phase of the cell cycle, because Clb5- and Clb2-dependent kinases are primarily responsible for phosphorylating origins of replication in the yeast cell genome and thereby triggering the onset of S phase. As long as $ORI(t) < 1$, the cell is in G1 phase; as soon as ORI(t) exceeds 1, the cell enters S phase. After that decision point is past, the model takes no further notice of the value of ORI(t). It can increase to any arbitrary value, because it is reset to 0 when the cell divides. So no penalty should be given for ORI(t) getting larger than 40 in a viable strain.

Response 1.6: Based on the results shown for the fit of Oguz et al. (2013) in Table S2, we believe our choices of upper bounds for ORI(t) and SPN(t) are acceptable. Drs. Tyson and Kraikivsky raise the concern that in simulations of viable mutants, ORI(t) might exceed our upper bound, but remain oscillatory and therefore consistent with a viable phenotype. We see that this particular situation -- a penalty for exceeding the ORI(t) upper bound in an otherwise acceptable viable case -- occurs infrequently when evaluating the fit of Oguz et al. (2013), and when it does occur, it results in a very small penalty. In Table S2, the total penalty resulting from this case is 3.27, with a maximum of 0.51 on a single mutant. We have added this analysis to the footnote of Table S1.

Of course, a G1-arrested cell should be penalized if $ORI(t)$ exceeds 1.0 (or 0.75, we won't quibble), but the penalty should not be outrageously high as in Table S2.

Response 1.7: Drs. Tyson and Kraikivsky's second concern about $ORI(t)$ is the weight we assigned to the constraint -- In cases where the fit of Oguz et al. (2013) failed the requirement that $ORI(t)$ must be oscillatory in viable cells, and must not reach 1 in cells in G1 arrest, the penalty resulting from the constraint violation is high.

Our choice of the weights of the $ORI(t)$ constraints was based on our goal to satisfy the spirit of the model: We want to prevent cases in which a parameter set produces incorrect model behavior, but still achieves a low objective function. Specifically, for cells in G1 arrest, $ORI(t)$ must not reach 1, and we need to impose a large enough penalty to prevent this behavior. As the constraint is written, a time course that reached exactly 1 would receive a penalty of 0.5, which is large enough to push the fitting away from such a parameter set. However, if we reduced the constraint weight by a factor of 10, the penalty of 0.05 would not be large enough -- we would be likely to get fits that allow $ORI(t)$ to increase above 1 and take the small penalty. An unavoidable consequence of the constraint weight is that parameter sets that fail the constraint by a large amount receive a large penalty.

Similar reasoning holds for our choices of weight for the $ORI(t)$ upper bound on viable cells: If $ORI(t)$ reaches ~500, it is likely increasing without bound rather than oscillating, and should receive a penalty, which implies that if $ORI(t)$ reaches ~50,000, the penalty will be very large.

We have added the above justification of choices of weights as the third footnote of Table S1.

In light of this motivation for constructing certain features of the objective function, Drs. Tyson and Kraikivsky are correct that it is unfair to claim that the fit in the present study is ~13 times better than that of Oguz et al. (2013) based on the value of the objective function. The more appropriate comparison is in the number of mutants with phenotypes correctly simulated using the best-fit parameter set. In the main text, we now avoid comparing the methods based on the value of the qualitative objective function, and instead point primarily to Fig. S3 as a means of comparison of the methods.

We think this problem needs to be corrected and the best-fits need to be recalculated for both optimization methods. As the procedure currently stands, the objective function overweights the ORI variable, which will force the optimization routine to choose parameter values that limit the growth of $ORI(t)$ after it crosses $ORI = 1$, even though the subsequent increase in ORI is inconsequential to the model.

Similar considerations apply to $BUD(t)$ and $SPN(t)$, which are also flags to determine the timing of bud emergence and spindle assembly.

Response 1.8: In summary, features of the objective function involving $ORI(t)$, $SPN(t)$, and $BUD(t)$ help to enforce the desired model properties in the best fit, by preventing

situations in which a parameter set could violate a known system property but still achieve a low objective value. We have made the revisions described in Responses 1.5-1.7, to make our motivation for these features of the objective function more clear to the reader.

A few minor problems should also be addressed:

(1) In viable cells, [C1b2] must drop below K_{EZ} as cells exit mitosis, in order to relicense origins of replication. We don't see that this constraint is included in Mitra's objective function.

Response 1.9: The property is included indirectly with the constraints $FLAG_UDNA(\tau_2) \geq 1$; τ_2 such that $DIV_COUNT(\tau_2) = 2$ and $FLAG_UDNA(\tau_3) \geq 1$; τ_3 such that $DIV_COUNT(\tau_3) = 3$.

As described in Response 1.10 below, $FLAG_UDNA$ is a variable that has a value of 0 if origin activation has not yet occurred in the current cell cycle, and changes to 1 at the time point when $ORI(t)$ crosses 1. $FLAG_UDNA$ resets to zero at cell division, but $ORI(t)$ is reset to zero only when origin relicensing occurs - we follow the model of Oguz et al. (2013) and use the condition $[C1b2] + [C1b5] < 0.2$, which is similar to Drs. Tyson and Kraikivsky's suggested criterion. If origin relicensing did not occur, then $ORI(t)$ would not cross 1 during the second cycle (it would remain above 1 for the entire time), so $FLAG_UDNA$ would remain equal to zero and the constraint $FLAG_UDNA(\tau_2) \geq 1$ would be failed.

We have updated the "System Property" description of these constraints in Table S1 to indicate that these constraints indirectly enforce origin relicensing.

(2) What is the meaning of the "FLAG" variables ($FLAG_BUD(t)$, $FLAG_UDNA(t)$, etc.) in Table S1?

Response 1.10: The "FLAG" variables are variables that can only take values of 0 or 1, used to track the occurrence of certain events during each cell cycle ($FLAG_BUD$ for bud formation, $FLAG_UDNA$ for origin activation, and $FLAG_SPN$ for spindle assembly). Each flag is reset to 0 at each cell division, and changes to 1 when the corresponding continuous variable crosses 1 (BUD for $FLAG_BUD$, ORI for $FLAG_UDNA$, SPN for $FLAG_SPN$). We have added these details to the text below Table S1.

(3) It would help many readers if the authors were to include a brief description of "enhanced scatter search" in the section on Materials and Methods.

Response 1.11: We briefly describe enhanced scatter search in the last paragraph of section 4.3. We have added more information to this paragraph for the benefit of readers unfamiliar with the algorithm, namely that scatter search is a type of evolutionary optimization algorithm.

In summary, we think that Mitra et al. have clearly described a new and valuable approach to parameter estimation in large, realistic, mathematical models of molecular regulatory mechanisms in systems biology. They have described a reasonable way to incorporate both qualitative and quantitative data into a single, scalar objective function, and they have explored the feasibility of a reasonable method (enhanced scatter search) to minimize the objective function. But the details of their calculations are seriously marred by a misunderstanding of the roles of the "flags" in the Oguz model. By overweighting the importance of these flags, they may have skewed their optimization routine into regions of parameter space that are not particularly relevant to the spirit of the mathematical model. This mistake seriously compromises the conclusions they draw from their calculations, especially the surprisingly poor performance of the Oguz model on the mutant-phenotype observations. We recommend that the objective function be modified to properly account for the "flags" and that the calculations be repeated with the new objective function.

Response 1.12: As discussed above, we expect that if the calculations were repeated with reduced or eliminated ORI(t) penalties, the fit would be less consistent with the desired system properties for ORI(t), because it would be possible to have a parameter set that fails to satisfy the system properties, but receives little or no penalty.

We hope we have addressed Drs. Tyson and Kraikivsky's concerns about the ORI(t) constraints by making our motivation for including them more clear in the Supplementary Information, and by being more careful in the comparisons we draw between our fit and the fit of Oguz et al (2013).

Reviewer #2 (Remarks to the Author):

Mitra et al. present a case study on using qualitative data for calibrating quantitative dynamical models. The manuscript also progresses toward a formalism for using qualitative data to calibrate models and provides practical advice on how to use qualitative data to calibrate models, including how to choose weights in error functions and algorithms for constrained optimization. The manuscript is a valuable contribution because it highlights an abundant source of data (qualitative data) that could, but is often not, used to calibrate models. This is particularly important for building larger models that represent each gene product that are needed to support medicine and bioengineering. The manuscript is well-written and well-organized.

Response 2.1: We are delighted that the reviewer found our manuscript to be a valuable contribution that is well-written and well-organized.

However, we feel that the manuscript could have more impact on the field with a few small improvements to the text.

Major comments

The manuscript could be strengthened by adding a brief section which presents a formal definition of the model calibration using quantitative and qualitative data. The manuscript does present a definition in Section 2.2.2. Nevertheless, it would be helpful for readers to present this separately from the individual case studies.

Response 2.2: We added the suggested formalization as a new Section 2.2.

The manuscript could be strengthened by adding a brief section with practical advice on how to use both quantitative and qualitative data for calibration. This could include recommendations for databases to obtain qualitative data,

constrained optimization algorithms, optimization packages. The manuscript does have recommendations throughout the text. Nevertheless, it would be helpful to organize these recommendations into a single section.

Response 2.3: We have added a section to the Discussion (Section 3.1) providing practical advice on combining qualitative and quantitative data for model fitting.

The manuscript could be strengthened by adding a brief exposition on the similarity of the approach presented to model checking/validation.

Response 2.4: We have added a paragraph to the introduction comparing the presented approach to model checking, and added references to :

- Clarke, E. M.; Grumberg, O.; Peled, D.; Belta, P. C. Model Checking; The Cyber-Physical Systems Series; MIT Press, 1999.
- Calzone, L.; Fages, F.; Soliman, S. BIOCHAM: An Environment for Modeling Biological Systems and Formalizing Experimental Knowledge. Bioinformatics 2006, 22 (14), 1805–1807.
- Clarke, E. M.; Faeder, J. R.; Langmead, C. J.; Harris, L. A.; Jha, S. K.; Legay, A. Statistical Model Checking in BioLab: Applications to the Automated Analysis of T-Cell Receptor Signaling Pathway. In Computational Methods in Systems Biology; Heiner, M., Uhrmacher, A. M., Eds.; Springer Berlin Heidelberg: Berlin, Heidelberg, 2008; pp 231–250.

Minor comments

Section 2.1 could be eliminated for brevity.

Response 2.5: Given that space permits, we have chosen to keep the section because we have found this material to be useful in talks.

Please clarify the qualitative error function $g_i(x)$ on page 4. Is this Boolean-valued or $\max\{0, A(l_i, x) - A(0)\}$?

Response 2.6: $g_i(x)$ is a scalar function used to define a constraint, $g_i(x) < 0$. This constraint is incorporated into the scalar error function as defined in Equation 3. We have made our definition of $g_i(x)$ more clear, now defining it in Section 2.2 separately from the Raf example.

It would be interesting to comment on why this error function, $g_i(x)$ was chosen instead of alternatives, such as the difference $\max\{0, A(l_i, x) - A(0)\}$ or squared difference $\max\{0, A(l_i, x) - A(0)\}^2$

Response 2.7: Note that our error function is equivalent to using the difference: the error function associated with the inequality constraint $g_i(x) < 0$ is $\max(0, g_i(x))$. What this means is if the constraint is satisfied, the error function is 0, and if not, the error function is proportional to the difference between the simulated value and the bound

specified by the constraint. In the particular example of the Raf model, $g_i(x) = A(l_i, x) - A(0)$, resulting in the form that the reviewer mentions.

The difference, $\max(0, g_i(x))$, is a common choice for the static penalty function in the optimization literature -- we have added a reference to Smith and Coit (1997) in Section 2.2 to justify this choice.

The squared difference $\max(0, g_i(x))^2$ is sometimes used as well. We added to Section 2.2 a justification of why we chose the difference: to avoid overpenalizing single constraints with a large degree of violation.

The second example is informative, but the evidence in Section 2.2.3 and Table 1 that using both quantitative and qualitative data outperforms using only quantitative or qualitative data is weak. Other examples may be able to more clearly demonstrate the utility.

Response 2.8: We find that the improvement in the bounds of K3 is considerable -- bounded to within 2 orders of magnitude with either individual dataset compared to 1.2 orders of magnitude with the combined datasets. We have added a specific note of this comparison in the text.

Of course, the level of improvement will depend on the number of data points and the level of noise in each of the synthetic datasets.

The comparison between using both quantitative and qualitative data to using only quantitative data in Section 2.3.2 lacks context. The demonstration would be bolstered by independently validating the revised parameter values with new experimental data.

Response 2.9: We have rewritten Section 2.3.2 (now numbered 2.4.2) to provide more context on how the fit should be evaluated, noting that the main criterion to evaluate agreement with qualitative data is that each mutant satisfies the system properties in Table S1.

We agree that it would be interesting to evaluate the fitted model on new data that were not included in the fit, but acquisition of such new data is outside the intended scope of the current study.

In Section 2.3, where the yeast cell cycle model is introduced, please clarify what data set/types were used for parameter identification in the original study. Currently, this information is not clear. This makes it harder to evaluate the comparison of model fits (Figure 4).

Response 2.10: We have added an appropriate description of the original study in section 2.4, paragraph 1:

“The previous fit [ref. 7] incorporated the phenotypes of 119 yeast strains (viable or inviable, but without considering the phase of cell cycle arrest). Nominal parameter values, based on earlier, hand-tuned models, were used to limit the search space of the optimization algorithm: The search covered a range of $\pm 90\%$ of each nominal value.”

Small legends would improve the readability of Figures 4 and 5 by enabling the reader to focus on the figure rather than reading the caption.

Response 2.11: We have added the suggested legends to the figures.

Reviewer #3 (Remarks to the Author):

Combining qualitative and quantitative data for parameter identification is a very valuable goal. The authors aim to reach this goal by adding a term to the standard (quantitative data) objective function that penalizes a violation of the qualitative data. They exemplify their approach on two simple simulated data sets and on real world data from the yeast cell cycle control.

While the proposed methods is intuitive, my main criticism is that it lacks any statistical rigour.

It starts with eq. (2) in which each term of the sum should be divided by the variance of the measurement error to turn it into a maximum likelihood estimator that reflects the confidence in the quantitative data. As the authors state correctly, C in eq. (3) reflects the confidence in the qualitative data.

Response 3.1:

The reviewer raises a valid point that a remaining challenge associated with this approach is an acceptable method of choosing C_i . This is a known limitation of static penalty function methods, as described in

Runarsson, T. P.; Yao, X. Stochastic Ranking for Constrained Evolutionary Optimization. *IEEE Trans. Evol. Comput.* 2000, 4 (3), 284–294.

We have added a paragraph to the Discussion section that describes the current state of the optimization field in regards to this issue:

“A known limitation of this method is the need to choose penalty constants C_i for each constraint [ref. 25]. Although progress has been made in developing constrained optimization algorithms that do not require hand-chosen constants [ref. 8], future work is required to adapt such algorithms to biological applications---in particular, available algorithms are not designed for problems such as that of the yeast cell cycle model, in which not every constraint is satisfied in the best fit.”

But the next sentence says: 'We hand-selected a value of $C=0.03$ to give roughly equal contributions from the qualitative and qualitative data sets'. There is no reason why these contributions should be equal.

Response 3.2: In Section 2.3 (old Section 2.2), we are analyzing an example with synthetic data, so there is no “correct” relative weighting of the two datasets -- the decision would depend on how much weight the modeler places in the two experiments. We have edited the text to clarify that making the qualitative and quantitative datasets contribute roughly equally is a choice that we made for the purposes of illustration, but is not the only possible choice.

The situation becomes worse if there are many C_i . Section 2.3.1: 'The C_i were hand-chosen ... such that each constraint contributed roughly equally.' Why should they ? One constraint might be very strict, another rather loose.

Response 3.3: In Section 2.4 (old Section 2.3), it was necessary to choose the C_i for each constraint, and we provide some justification for our choices. First, we point out that it is necessary to adjust the C_i based on the order of magnitude of the variables. For example, if one constraint can be violated by a value of order 100 and another can be violated by a value of order 1, the latter should have a larger weight. Otherwise, the fitting would focus solely on the constraints with large absolute magnitudes of violation.

Second, as the reviewer points out, it is possible to have some constraints that are more strict than others, and this level of strictness could be enforced by changing the constraint weight. For the most part, we did not have enough information to make such a judgement in this model, and so used the C_i only for the order-of-magnitude adjustment described above. However, we note that for constraints on wild type cells, we do have more information about strictness (we are certain that wild type cells are viable, so the constraints should be very strict), and we increased the corresponding C_i accordingly (described in Section 2.4.1).

We have added Section 3.1 to the Discussion, with practical advice on performing fitting that incorporates both qualitative and quantitative data. We included in this section the above suggestions on how one should go about choosing the C_i .

Last paragraph of Sec. 2.3.1 'This factor [C_i scaling] was tuned by trial and error ...'

Response 3.4: In section 2.3.1 we state that one value was tuned by trial and error - a scaling factor of 1/15 applied to all of the C_i 's (the C_i themselves were not tuned, but chosen as described above), such that the fit achieved agreement with both the qualitative and quantitative data.

The procedure would be more automated if the need for a hand-tuning step could be eliminated in future work, but in the current study, tuning of one scaling factor is a significant improvement over tuning 150 model parameters.

As part of the new Section 3.1, we included a useful heuristic to set the scaling factor for all C_i (1/15 in our example), based on the best-fit values of f_{quant} and f_{qual} , to reduce the need for trial-and-error hand-tuning.

In conclusion, the proposed methods replaces the hand-tuning of parameters by Novak & Tyson to come up with their first cell cycle model by hand-tuning of C_i 's.

Response 3.5: Although the need for choices of C_i remains a limitation of the approach, by following the guidelines described in Section 3.1, it is possible to choose the C_i in a logical way without the need for iterative hand-tuning.

REVIEWERS' COMMENTS:

Reviewer #1 (Remarks to the Author):

Mitra ... Hlavacek have responded at length to all our comments and suggestions on their paper, and, although we do not agree with all of their responses, we agree that the revised paper treats the budding yeast model with more clarity and makes a fairer comparison of their parameter fitting approach to that of Oguz et al. Hence, we are willing to recommend the revised version for publication in Nature Communications.

Our remaining misgivings concern the authors' responses 1.5 and 1.6.

Response 1.5: Drs. Tyson and Kraikivsky are correct to point out that in the context of the model, $ORI(t)$, $BUD(t)$, and $SPN(t)$ only have meaning in terms of when they cross values of 1, and so in principle, it is acceptable for them to increase to an arbitrarily large value. However, it is required for these variables to have oscillatory behavior in viable cells, with a local maximum in each cell cycle. As a surrogate for "has oscillatory behavior", which can be difficult to express in terms of inequality constraints, we chose to impose upper bounds on each of the variables - this proved effective because the alternative to oscillatory behavior in this model is for the variable to increase without bound ...

Response 1.6: Based on the results shown for the fit of Oguz et al. (2013) in Table S2, we believe our choices of upper bounds for $ORI(t)$ and $SPN(t)$ are acceptable. Drs. Tyson and Kraikivsky raise the concern that in simulations of viable mutants, $ORI(t)$ might exceed our upper bound, but remain oscillatory and therefore consistent with a viable phenotype ...

Our reply: These variables always "oscillate" in viable cells because they are reset to 0 at cell division and then increase above 1 at some point in the next simulated cycle before being reset to 0 at the next division. They may or may not have a "local maximum" in each cell cycle. It is possible (and perfectly acceptable, as far as the model is concerned) that one or more of these "flags" increases monotonely to a large value (> 40) and then gets abruptly reset to 0 at cell division. The model pays no attention to these flags, once they exceed "1" (which only indicates that a certain event has been executed). The authors' insistence that these variables have a local maximum < 40 in each cell cycle are additional constraints that they impose, which are outside the "spirit" of the original model. We allow them these additional constraints—it is their paper, not ours—because, in the revised version, they do not "overplay" these cards by making unfair comparisons of their "fit" to the data to the "fit" of Oguz et al.

Similar considerations apply to the penalty assigned to $ORI(t)$ in inviable cells of a mutant strain blocked in G1. If $ORI(t)$ exceeds 0.75 in a simulation of a G1-blocked cell, then the mutant phenotype is incorrectly simulated and should be punished, but it should not be punished more and more severely as $ORI(t)$ continues to increase to a very large value.

The problem with the large penalties that the authors impose on ORI is that the penalties skew the optimization procedure to parameter values that try to keep ORI under control at the expense (perhaps) of other more important constraints on the model. For example, the Oguz et al. parameter set, although it gives a perfectly good fit to the qualitative and quantitative data, would be soundly rejected by the optimization procedure because of the large penalties incurred because $ORI(t)$ does not conform to the arbitrary constraints applied by Mitra et al. This is analogous to throwing a poor ghetto kid into jail for ten years for possessing a few ounces of marijuana while the

Masters of Wall Street go scot-free for ruining the global economy.

Whereas we would have preferred the authors to repeat their calculations with a more reasonable objective function, we will not insist on our preferred approach. In the end, the authors propose an interesting approach to an important problem in molecular systems biology, and they now clearly explain exactly how and why they are carrying out the calculations as they did. We may not agree with all their decisions, but the paper is now OK for publication.

John Tyson and Pavel Kraikivsky

Reviewer #2 (Remarks to the Author):

We commend the authors for satisfying our concerns.

In our opinion, modeling must be systemized and scaled to enable more comprehensive and more predictive models. We think that this article helps systemize a common, but typically informal practice in modeling, and that this systemization will help enable better models. We think that the article is also approachable and provides simple, practical advice for modelers. For these reasons, we believe this article would be a valuable contribution.

REVIEWERS' COMMENTS:

Reviewer #1 (Remarks to the Author):

Mitra ... Hlavacek have responded at length to all our comments and suggestions on their paper, and, although we do not agree with all of their responses, we agree that the revised paper treats the budding yeast model with more clarity and makes a fairer comparison of their parameter fitting approach to that of Oguz et al. Hence, we are willing to recommend the revised version for publication in Nature Communications.

Our remaining misgivings concern the authors' responses 1.5 and 1.6.

Response 1.5: Drs. Tyson and Kraikivsky are correct to point out that in the context of the model, $ORI(t)$, $BUD(t)$, and $SPN(t)$ only have meaning in terms of when they cross values of 1, and so in principle, it is acceptable for them to increase to an arbitrarily large value. However, it is required for these variables to have oscillatory behavior in viable cells, with a local maximum in each cell cycle. As a surrogate for "has oscillatory behavior", which can be difficult to express in terms of inequality constraints, we chose to impose upper bounds on each of the variables - this proved effective because the alternative to oscillatory behavior in this model is for the variable to increase without bound ...

Response 1.6: Based on the results shown for the fit of Oguz et al. (2013) in Table S2, we believe our choices of upper bounds for $ORI(t)$ and $SPN(t)$ are acceptable. Drs. Tyson and Kraikivsky raise the concern that in simulations of viable mutants, $ORI(t)$ might exceed our upper bound, but remain oscillatory and therefore consistent with a viable phenotype ...

Our reply: These variables always "oscillate" in viable cells because they are reset to 0 at cell division and then increase above 1 at some point in the next simulated cycle before being reset to 0 at the next division. They may or may not have a "local maximum" in each cell cycle. It is possible (and perfectly acceptable, as far as the model is concerned) that one or more of these "flags" increases monotonely to a large value (> 40) and then gets abruptly reset to 0 at cell division. The model pays no attention to these flags, once they exceed "1" (which only indicates that a certain event has been executed). The authors' insistence that these variables have a local maximum < 40 in each cell cycle are additional constraints that they impose, which are outside the "spirit" of the original model. We allow them these additional constraints—it is their paper, not ours—because, in the revised version, they do not "overplay" these cards by making unfair comparisons of their "fit" to the data to the "fit" of Oguz et al.

Similar considerations apply to the penalty assigned to $ORI(t)$ in inviable cells of a mutant strain blocked in G1. If $ORI(t)$ exceeds 0.75 in a simulation of a G1-blocked cell, then the mutant phenotype is incorrectly simulated and should be punished, but it should not be punished more and more severely as $ORI(t)$ continues to increase to a very large value.

The problem with the large penalties that the authors impose on ORI is that the penalties skew the optimization procedure to parameter values that try to keep ORI under control at the expense (perhaps) of other more important constraints on the model. For example, the Oguz et al. parameter set, although it gives a perfectly good fit to the qualitative and quantitative data, would be soundly rejected by the optimization procedure because of the large penalties incurred because $ORI(t)$ does not conform to the arbitrary constraints applied by Mitra et al. This is analogous to throwing a poor ghetto kid into jail for ten years for possessing a few ounces of marijuana while the Masters of Wall Street go scot-free for

ruining the global economy.

Whereas we would have preferred the authors to repeat their calculations with a more reasonable objective function, we will not insist on our preferred approach. In the end, the authors propose an interesting approach to an important problem in molecular systems biology, and they now clearly explain exactly how and why they are carrying out the calculations as they did. We may not agree with all their decisions, but the paper is now OK for publication.

John Tyson and Pavel Kraikivsky

Response: We are grateful for the positive recommendation from Drs. Tyson and Kraikivski, and we understand their remaining concerns. We offer a response here to reveal more of our thinking about these concerns. The concerns revolve around certain aspects of our definition of the objective function to be minimized, namely the penalty terms and related constraints imposed on one of the model variables, $ORI(t)$, which serves the purpose of tracking the progress/completion of origin activation, the details of which are not explicitly considered in the yeast cell cycle model (developed and refined by Tyson and co-workers over many years). When $ORI(t)$ is greater than 1 in value, the process represented by the variable is taken to be complete. We imposed the following constraints on this variable for the reasons explained in manuscript: $ORI < 40$ for viable strains, and $ORI < 0.75$ for mutant strains in G1 arrest. Although we found this approach to be helpful (in steering the search of parameter space away from parameter values that yield pathological behavior) and workable, in that we obtained parameter estimates that allow the model to reproduce the qualitative and quantitative observations available, we would not be surprised if someone found a better or more elegant way to formulate the ORI-specific aspects of the objective function. We recognize and agree that the current formulation has the potential to penalize some $ORI(t)$ behavior that is not by itself pathological. At the end of the day, these issues are model-specific and may not arise in other applications, and modeler discretion will probably always play some role in the formalization of data-driven constraints on model-predicted behaviors. The goal of a constraint or set of constraints is to enforce agreement with qualitative observations. It seems likely that a given goal might be achievable through multiple, distinct formulations of constraints. We don't have a lot of experience with this approach to parameter identification at this time, and it might be very interesting to evaluate alternative approaches to constraint formulation in future work.

Reviewer #2 (Remarks to the Author):

We commend the authors for satisfying our concerns.

In our opinion, modeling must be systemized and scaled to enable more comprehensive and more predictive models. We think that this article helps systemize a common, but typically informal practice in modeling, and that this systemization will help enable better models. We think that the article is also approachable and provides simple, practical advice for modelers. For these reasons, we believe this article would be a valuable contribution.

Response: We are delighted that the reviewer is satisfied with our revisions and found our article to be a valuable contribution.